# Differential Association of the *DISC1* Interactome in Hallucinations and Delusions [note 1]

**DOI:** 10.3390/ijms26178738

**Published:** 2025-09-08

**Authors:** Araceli Gutiérrez-Rodríguez, Alma Delia Genis-Mendoza, Jorge Ameth Villatoro-Velázquez, María Elena Medina-Mora, Humberto Nicolini

**Affiliations:** 1Laboratorio de Genómica de Enfermedades Psiquiátricas y Neurodegenerativas, Instituto Nacional de Medicina Genómica, Ciudad de México 14610, Mexico; araceligr1623@gmail.com (A.G.-R.); adgenis@inmegen.gob.mx (A.D.G.-M.); 2Posgrado en Ciencias Biológicas, Unidad de Posgrado, Edificio D, 1° Piso, Circuito de Posgrados, Ciudad Universitaria, Alcaldía Coyoacán, Ciudad de México 04510, Mexico; 3Servicios de Atención Psiquiátrica, Hospital Psiquiátrico Infantil “Juan N. Navarro”, Ciudad de México 14080, Mexico; 4Unidad de Análisis de Datos y Encuestas, Instituto Nacional de Psiquiatría Ramón de la Fuente Muñiz, Ciudad de México 14370, Mexico; ameth@imp.edu.mx (J.A.V.-V.); metmmora@gmail.com (M.E.M.-M.)

**Keywords:** delusions, *DISC1*, genetic association, hallucinations, interactome

## Abstract

Multiple genes within the *DISC1* (Disrupted-in-Schizophrenia-1) interactome have been implicated in psychotic disorders, which are characterized by hallucinations, delusions, negative symptoms, and disorganized behavior. However, the genetic associations of specific psychotic symptoms remain poorly understood. Methods: We conducted a genetic association analysis of the *DISC1* interactome for hallucinations and delusions in schizophrenia and bipolar disorder, using single-nucleotide polymorphism (SNP), gene, and gene-set approaches. Results: Our findings showed an association between the SNP rs6754640 in the *NRXN1* gene and auditory hallucinations. Additionally, rs10263196 (*EXOC4*), rs7076156 (ZNF365), and nine *NRXN1* SNPs were associated with delusions of reference, while rs17039676 (*NRXN1*) was linked to persecutory delusions. At the gene level, *NRG1* and *PCM1* were related to auditory hallucinations. The *NRXN1*, *APP*, *EXOC4*, and *NUP210* genes were associated with delusions of reference, whereas *NRG1* and *APP* were linked to persecutory delusions. Gene-set analysis indicated that pathways related to the regulation of neuronal structure and function were involved in auditory hallucinations, while cellular transport regulation pathways were associated with persecutory delusions. Conclusions: This study emphasizes the polygenic architecture of psychosis and suggests that distinct molecular mechanisms contribute to different types of hallucinations and delusions.

## 1. Introduction

Genetic variants within the *DISC1* (Disrupted-in-Schizophrenia-1) interactome have been associated with psychosis [1]. The *DISC1* interactome comprises a network of over 150 proteins, with the DISC1 protein serving as a scaffold [2,3]. Enriched interaction analyses of both common [4] and rare genetic variants [2] within the DISC1 interactome, conducted through gene set analysis, have been associated mainly with schizophrenia (SCZ) [2,3,4], a disorder characterized by positive symptoms (psychosis) and negative symptoms (reduced emotional expression), which leads to a decline in both individual and social well-being [5]. Psychosis is also a highly prevalent phenotype in patients with bipolar disorder (BD) who not only exhibit affective symptoms [6]. BD is a condition that partially overlaps both clinically and genetically with schizophrenia [7]. This disabling phenotype is primarily characterized by hallucinations and delusions, reflecting an inability to distinguish internal fantasies from external reality [8,9].

Psychosis exhibits high heritability (~80%), and genome-wide association studies (GWAS) of psychotic disorders, including SCZ and BD, have identified partially overlapping risk loci [1,7]. Moreover, polygenic risk analyses indicate that genetic correlations support a shared genetic architecture of psychosis across both SCZ and BD [10]. Despite the heterogeneity of psychosis, due to the diverse presentation of symptoms and genetic background, most genetic association studies have focused on the overall phenotype [11,12,13] or the psychotic disorder itself [8,14,15,16]. Only a few have examined specific psychotic symptoms [17,18,19,20], resulting in limited evidence regarding the genes or pathways underlying individual symptoms. Such knowledge could enhance our understanding of the biological complexity of psychosis. We hypothesize that genetic variants within the *DISC1* interactome are differentially associated with specific positive psychotic symptoms, regardless of the overall psychotic disorder diagnosis. To our knowledge, this is the first report to associate various types of hallucinations and delusions with genetic variants, genes, or gene sets at risk loci for psychosis. This study aims to examine the associations between *DISC1* interactome genes and specific hallucinations (auditory, visual, somatic, olfactory) and delusions (guilt, reference, grandiose, persecutory, thought control) in individuals with schizophrenia and bipolar disorder in the Mexican population.

## 2. Results

### 2.1. Genetic Association with Hallucinations and Delusions at the Single Nucleotide Polymorphism (SNP) Level

A nominal association was identified between the SNP rs6754640 in the Neurexin 1 (*NRXN1*) gene and auditory hallucinations (*p* = 1.48 × 10^−5^, OR = 2.27). No statistically significant associations were observed for visual, somatic, or olfactory hallucinations. For delusions of reference, several SNPs within *NRXN1* were nominally associated, including rs6706713 (*p* = 6.37 × 10^−7^, OR = 3.40), rs11892200 (*p* = 7.05 × 10^−7^, OR = 3.15), rs6754640 (*p* = 2.54 × 10^−6^, OR = 3.33), rs17039676 (*p* = 3.52 × 10^−6^, OR = 4.13), rs6731061 (*p* = 3.95 × 10^−6^, OR = 3.05), rs7578902 (*p* = 4.11 × 10^−6^, OR = 2.93), rs10189159 (*p* = 7.97 × 10^−6^, OR = 3.14), rs10176705 (*p* = 2.64 × 10^−5^, OR = 3.11), and rs1421579 (*p* = 2.83 × 10^−5^, OR = 2.52).

Additional nominal associations for delusions of reference were detected with rs7076156 in the Zinc Finger Protein 365 (*ZNF365*) gene (*p* = 1.13 × 10^−5^, OR = 3.48) and rs10263196 in the Exocyst Complex Component 4 (*EXOC4*) gene (*p* = 1.36 × 10^−5^, OR = 3.09). In persecutory delusions, we again identified the SNP rs17039676 in the *NRXN1* gene (*p* = 8.43 × 10^−6^, OR = 4.89). SNP-level analyses did not show statistically significant associations for delusions of guilt, grandioseness, or thought control (Table 1).

According to their functional consequences, most of the identified SNPs in *NRXN1* were intronic or downstream variants, frequently associated with nonsense-mediated decay in protein-coding transcripts. For the majority of SNPs, their allele frequencies differed from those reported in the GnomAD database. The SNP rs7076156 in *ZNF365* was observed to affect a non-coding transcript, while the SNP in *EXOC4* was located in an intronic region of a protein-coding gene (Table 2).

### 2.2. Genetic Associations with Hallucinations and Delusions at the Gene Level

We observed an association between Neuregulin 1 (*NRG1*) and auditory hallucinations in patients with psychosis (*p* = 5.25 × 10^−6^, Z = 4.40), as well as between the gene encoding pericentriolar material 1 (*PCM1*) and auditory hallucinations (*p* = 4.47 × 10^−4^, Z = 3.32). No significant associations were found for visual, olfactory, or somatic hallucinations.

For delusions of reference, significant associations were identified with *NRXN1* (*p* = 7.57 × 10^−6^, Z = 4.33) and *EXOC4* (*p* = 3.69 × 10^−4^, Z = 3.37), as well as with *NUP210*, encoding nucleoporin 210 (*p* = 1.90 × 10^−4^, Z = 3.55), and *APP*, encoding amyloid precursor protein (*p* = 2.52 × 10^−4^, Z = 3.48). In the analysis of persecutory delusions, associations were also observed with *NRG1* (*p* = 6.35 × 10^−5^, Z = 3.83) and *APP* (*p* = 5.34 × 10^−4^, Z = 3.27), both of which were also implicated in delusions of reference. No statistically significant gene-level associations were found for delusions of guilt, grandiose, or thought control (Table 1).

### 2.3. Genetic Association with Hallucinations and Delusions at the Gene Set Level

A gene set analysis was conducted to identify clusters of genes that may play a role in specific pathways associated with hallucinations and delusions. Pathway enrichment analyses were run using Gene Ontology (GO) and the Kyoto Encyclopedia of Genes and Genomes (KEGG); significant associations were observed only with GO, while no differences were found in KEGG. Significant associations were found between auditory hallucinations and the following processes: non-motile cilia assembly (*p* = 8.01 × 10^−3^, BETA = 1.70), cilia organization (*p* = 1.23 × 10^−2^, BETA = 1.21), positive regulation of cellular component organization (*p* = 1.58 × 10^−2^, BETA = 1.50), biogenesis of cellular components (*p* = 4.88 × 10^−2^, BETA = 1.72), glutamate receptor signaling pathway (*p* = 2.36 × 10^−2^, BETA = 1.72), regulation of myeloid fibril formation (*p* = 3.28 × 10^−2^, BETA = 1.67), secretion (*p* = 3.30 × 10^−2^, BETA = 1.96), and microglia activation (*p* = 4.06 × 10^−2^, BETA = 1.67). We also observed a statistically significant association between persecutory delusions and the processes of vesicular Golgi transport (*p* = 1.95 × 10^−4^, BETA = 2.30) and neural development (*p* = 5.86 × 10^−4^, BETA = 2.08). Contrary to expectations based on the findings in SNPs and genes, no genetic association was found with any process related to the delusions of reference (Table 3). The results of the pathway enrichment analyses are illustrated in Figure 1, whereas Figure 2 shows the network of biological pathways and genes associated with psychotic symptoms, focusing on DISC1 interactome genes grouped into GO clusters.

## 3. Discussion

In this study, we identified genetic variants, genes, and gene sets associated with frequent psychotic symptoms in Mexican patients with SCZ and BD, including auditory hallucinations, persecutory delusions, and delusions of reference. The latter was the most prevalent. Auditory hallucinations were the most frequent symptom, occurring in over 80% of cases, followed by reference delusions in nearly 70%. In contrast, Colijn and Ismail carried out a meta-analysis reporting that visual hallucinations were the most common symptom, along with persecutory delusions [21]. These populations are primarily Caucasian, which may explain the differences in symptom frequencies observed in our population.

Evidence indicates a partial genetic overlap between schizophrenia and bipolar disorder, with a genetic correlation of approximately 0.6. Common variants identified through GWAS, a widely used method in genetics research that links specific genetic variants to certain diseases, contribute modestly to risk individually but cumulatively increase susceptibility. In contrast, some variants remain disorder-specific [7]. At the SNP level, we found associations with delusions, including several *NRXN1* variants and SNPs rs7076156 and rs10263196 in *ZNF365* and *EXOC4*, respectively. Most of the identified SNPs were intronic or non-coding, suggesting regulatory rather than protein-disrupting effects. Variants in *NRXN1*, *ZNF365*, and *EXOC4* may influence transcript stability, splicing, or gene regulation. Furthermore, their allele frequencies differ from those in GnomAD, suggesting possible population-specific contributions to psychiatric risk.

Gene-level analysis revealed associations between *NRXN1*, *EXOC4*, *NUP210*, and *APP* and the delusions of reference. These genes may function as potential biomarkers, a notion supported by Hill et al., who identified genes associated with hallucinations, including Protein Phosphatase 3 Catalytic Subunit Beta (*PPP3CB*) and Discs Large Homolog 1 (*DLG1*), or with delusions, such as Phosphodiesterase 4D (*PDE4D*), which is also part of the *DISC1* interactome [22]. However, in our study, we did not observe any association between *PDE4D* and psychotic symptoms.

Psychosis is influenced by multiple factors, making the identification of underlying biological pathways a challenging task. Gene set analysis enables the investigation of specific clusters of genes. Chaumette et al. [11] reported that hallucinations and delusions in psychosis are linked to dopaminergic and glutamatergic pathways and allow for the identification of endophenotypes that, in turn, confer stability to the psychotic phenotype. They also highlighted the role of N-methyl-D-aspartate (NMDA) receptor activity in psychotic dysfunction. The glutamatergic signaling pathway has been implicated in the severity of psychotic symptoms and the risk of hospitalization, particularly through the Rap1 GTPase signaling pathway [22]. Moreover, a pleiotropy-informed genome-wide analysis demonstrated that metabotropic glutamate receptor signaling exerts concordant effects in both SCZ and BD [23]. In our population, no serotonergic genes were associated with hallucinations or other psychotic symptoms, contrasting with Rivero et al., who reported associations between the serotonin transporter gene (*SLC6A4*) SNPs and auditory hallucinations [17].

At the gene set level, persecutory delusions were associated with vesicular Golgi transport and neuronal development. Within these processes, *DTNBP1*, previously linked to hallucinations in SCZ, acts through dopaminergic and glutamatergic pathways [18]. Cheah et al. identified associations of the SNP rs4236167 with auditory hallucinations and rs9370822 with both visual and auditory hallucinations [14]. In our study, glutamatergic neurotransmitter pathways were associated with auditory hallucinations, including the *APP* and *GRIA2* genes. *APP* has not been previously reported in relation to psychosis. In contrast, a study of Alzheimer’s disease found no association between *APP* and psychotic symptoms, suggesting that psychosis in Alzheimer’s may arise from neurodegenerative processes in the later stages of the disease [24]. Similarly, a recent case reported a patient with neurodegeneration who carried a p.Pro380Arg mutation in the *GIGYF2* gene, which encodes Grb-10 interacting GYF protein-2, as well as a duplication in the 22q11.2 region, and who presented with psychosis and early-onset dementia. Various genetic variants of *GIGYF2* have been implicated in schizophrenia and neurodegenerative diseases such as Parkinson’s disease and dementia with Lewy bodies [25]. *GRIA2*, in turn, has been widely implicated in psychotic disorders, including SCZ and BD, and its expression is reduced by certain antipsychotic treatments [26]. Although Crisafulli et al. reported no association between SCZ and *GRIA1*, *GRIA2*, or *GRIA4*, they observed that the *GRIA1* SNP rs381329 was linked to a lower severity of psychotic symptoms [26]. Another gene encoding a glutamate receptor subunit, *GRIN2A*, has been associated with schizophrenia through fine-mapping and functional genomic analyses. *GRIN2A* is also implicated in neurodevelopmental disorders and in critical neuronal processes, including synapse formation, highlighting the role of disrupted neuronal communication in this psychotic disorder [27].

We also identified a genetic association between *DISC1* and auditory hallucinations, but only at the gene set level, specifically within the non-motile cilia assembly pathway. Cilia play a key role in cell signaling, particularly in neurons [8], and alterations in their function have been reported in several psychiatric disorders, including MDD, BP, and SCZ [28]. Studies of the *DISC1* rs821616 (Ser704Cys) variant in patients experiencing a first psychotic episode showed higher scores on the Scale for the Assessment of Positive Symptoms (SAPS) and the Scale for the Assessment of Negative Symptoms (SANS) among those with hallucinations. Moreover, approximately 5% of the variance in hallucinations among patients with SCZ was associated with the *DISC1* Leu607Phe polymorphism [29].

*NRG1* is another gene associated with auditory hallucinations and persecutory delusions. *NRG1* and *ERBB4*, which encode the receptor tyrosine-protein kinase erbB-4, have been extensively studied in relation to SCZ and are involved in the negative regulation of the glutamatergic pathway via the N-methyl-D-aspartate receptor (NMDAR) [30]. Gene expression studies of NRG1 have reported increased expression in individuals with poorer functional outcomes, suggesting a role in the severity of psychotic symptoms [13]. Additionally, *NRG1* has been identified in GWAS and MRI studies, with the SNP rs12467877 showing a significant association with lateral ventricle enlargement, a hallmark feature of schizophrenia. Another gene associated with auditory hallucinations is *PCM1*, which encodes a protein that forms a complex with other members of the *DISC1* interactome, including *BBS4* and *DISC1*. Suppression of these genes disrupts neuronal migration, and mutations in PCM1 have been reported in families affected by psychosis [31].

Regarding neurodevelopmental alterations, we observed the involvement of vesicular transport, with several genes from the *DISC1* interactome, such as *TRIO* and *NRXN1*, playing a key role in the clustering process [32]. We also identified the *GSK3B* gene as being associated with auditory hallucinations through gene set analysis, particularly in the regulation of biogenesis. Previous studies of *GSK3B* polymorphisms (-1727A/T and -50C/T) have reported a relationship with the age of symptom onset, especially in TT homozygotes, who developed symptoms later; however, no association was found with the development of BD [33]. This suggests a differential genetic contribution across psychotic disorders and may explain the lack of significant associations at the SNP or gene level for certain symptoms.

Although *NRXN1* and its multiple SNPs are associated with both auditory hallucinations and persecutory/reference delusions, its involvement in the positive regulation of the cellular component organization pathway was observed only for auditory hallucinations. A similar pattern was noted for *EXOC4* and *NUP210*. While gene-level analysis and the SNPs of *EXOC4* showed an association with reference delusions, *EXOC4* was also implicated in Golgi vesicular transport alterations related to persecutory delusions. In contrast, NUP210 was not involved in any biological pathway. These findings can be explained by the distinction between gene-set analysis, which identifies enriched biological pathways containing associated genes, and gene-level analysis, which evaluates each gene independently by aggregating SNP-based statistics. This suggests that *NRXN1*, *EXOC4*, and *NUP210* may be individually associated with psychotic symptoms, but they do not appear to play a central role in the biological pathways underlying hallucinations and delusions.

SCZ and BD are psychotic disorders that share genetic risk regions associated with psychosis and can present with overlapping symptoms [15,34]. Furthermore, GWAS have identified shared genetic risk factors for SCZ and BD that affect brain structure at multiple levels, including reduced connectivity in parietal and posterior cingulate circuits [34]. Risk variants in cytokine-mediated inflammatory pathways, such as the *IL6R* gene polymorphism rs2228145, which encodes the interleukin-6 receptor, have been associated with both the susceptibility and the severity of psychotic symptoms in SCZ and BD, potentially impacting neurotransmission [35]. However, the genetic basis of hallucinations and delusions remains poorly understood. Previous studies in this area have generally aggregated symptoms rather than analyzing them by specific types, and consistent significant associations have not been identified [16,19]. Clinical studies involving multiple cohorts of patients diagnosed with SCZ have shown that persecutory delusions are the most common subtype [36]. To our knowledge, the only study investigating genetic associations with persecutory delusions was conducted in a Romanian population. This study reported shared risk loci between SCZ and BD and observed a trend of an association between SNPs rs3916971, rs778293, and rs1421292 in the *M24* gene and persecutory delusions [20]. The present study did not consider factors that may influence the occurrence of hallucinations and/or delusions, such as adverse childhood experiences, which have been linked to the development of hallucinations [37,38]. This represents a limitation that should be addressed in future research.

In summary, our study identified specific *DISC1* interactome variants, genes, and gene-sets associated with specific psychotic symptoms. *NRXN1*, *NRG1*, and *PCM1* were associated with auditory hallucinations, whereas *NRXN1*, *APP*, *EXOC4*, and *NUP210* were linked to delusions of reference, and *NRG1* and *APP* to persecutory delusions. Pathway analysis highlighted neuronal structure and function in auditory hallucinations and cellular transport in persecutory delusions. These findings suggest a polygenic and pleiotropic architecture of psychotic symptoms and reveal molecular pathways that may be involved in hallucinations and delusions in schizophrenia and bipolar disorder, potentially contributing to the development of precision medicine approaches.

## 4. Materials and Methods

### 4.1. Study Design

DNA samples from cases diagnosed with psychotic disorders (SCZ and BD, *n* = 237) were analyzed, including 85 females and 127 males, with a mean age of 40.35 ± 13.05 years. Mental health specialists performed a diagnostic evaluation based on DSM-5 criteria, which require the presence of two or more of the following symptoms: delusions, hallucinations, disorganized speech, grossly disorganized or catatonic behavior, and negative symptoms, for a significant portion of time over one month, independent of substance use. Control samples (*n* = 986) were obtained from the Mexican Genomic Database for Addiction Research (MxGDAR/ENCODAT), a representative population-based cohort comprising 202 men and 807 women, with a mean age of 40.82 ± 12.60 years. Among patients, the prevalence of hallucinations was as follows: 81% experienced auditory hallucinations, 29.31% had visual hallucinations, 12.93% experienced somatic hallucinations, and 5.17% had olfactory hallucinations. For delusions, the distribution was as follows: 18.50% with delusions of guilt, 69.83% with delusions of reference, 38.79% with grandiose delusions, 49.14% with persecutory delusions, and 36.20% with thought control delusions.

### 4.2. Genotyping and Quality Control

Genotyping of all samples was performed using the Infinium^®^ PsychArray 24 BeadChip microarray platform (Illumina, San Diego, CA, USA). SNPs corresponding to 91 *DISC1* interactome genes were extracted according to their genomic positions based on the human reference assembly GRCh37/hg19. Quality control (QC) procedures were applied, excluding SNPs and individuals with missing data greater than 2 × 10^−2^, SNPs with minor allele frequency (MAF) < 0.05, and SNPs deviating from Hardy–Weinberg equilibrium (*p* < 1 × 10^−6^). Principal component analysis (PCA) was performed to detect population stratification and to exclude related or genetically similar individuals, minimizing potential confounding due to ancestry.

### 4.3. Single Nucleotide Polymorphism (SNP)-Level Association Analysis

SNP-level association analyses were conducted using PLINK (v1.9). Logistic regression models were applied to test the association of each SNP with psychotic phenotypes. Covariates included biological sex at birth, age, the first five ancestry principal components from PCA, and substance use (alcohol, tobacco, and other psychoactive substances, including marijuana, cocaine, hallucinogens, methamphetamine, heroin, or inhalants). Odds ratios (ORs) with 95% confidence intervals (CIs) were calculated to quantify effect sizes. This adjustment accounted for potential gene-environment interactions and controlled for substance-induced psychosis, as all controls reported no substance use. The Variant Effect Predictor (VEP) database from Ensembl was used to evaluate the potential functional consequences of the genetic variants.

### 4.4. Gene-Level and Gene-Set Enrichment Analysis

Gene-level and gene-set association analyses were conducted using MAGMA (v1.10). SNPs passing QC were annotated to their corresponding DISC1 interactome genes. Gene-based association statistics were computed using the SNP-wise Mean and SNP-wise Top 1 models, which account for linkage disequilibrium (LD) between SNPs to provide aggregated gene-level effects. Gene-set analyses were conducted using MAGMA by clustering genes according to functional categories defined by the Gene Ontology (GO) and the Kyoto Encyclopedia of Genes and Genomes (KEGG) databases, two of the most widely used resources for molecular and biological functional annotation. These analyses tested whether the gene sets collectively exhibited a significant association with psychotic phenotypes.

### 4.5. Statistical Analysis

To correct for multiple comparisons across SNPs, genes, and gene sets, the False Discovery Rate (FDR) method was applied using the R statistical environment (v4.3.0). All analyses were evaluated at a nominal significance level of *p* < 0.05, and FDR-adjusted *p*-values were reported to control for type I error.

## Figures and Tables

**Figure 1 ijms-26-08738-f001:**
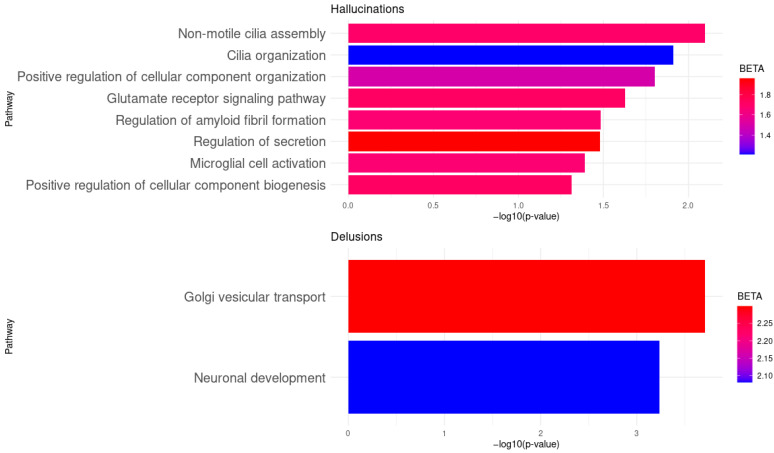
Gene Ontology-based enrichment of biological pathways associated with psychotic symptoms. Significantly enriched pathways are shown for hallucinations (top panel) and delusions (bottom panel). Pathway significance is expressed as –log_10_(*p*-value), and effect size is represented by BETA (red tones indicate higher values, blue tones lower values). Only pathways with *p* < 0.05 are included.

**Figure 2 ijms-26-08738-f002:**
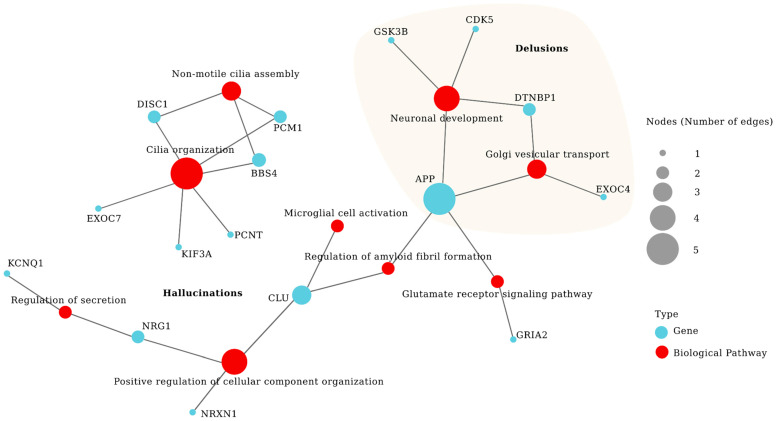
Network of biological pathways and genes associated with psychotic symptoms. Pathway enrichment analyses using MAGMA are presented, focusing on DISC1 interactome genes grouped into Gene Ontology (GO) clusters. Red circles represent biological pathways significantly associated with auditory hallucinations and persecutory delusions. Blue circles represent individual genes within each pathway. Circle size reflects the number of connections between genes and pathways, highlighting shared biological mechanisms.

**Table 1 ijms-26-08738-t001:** Genetic associations with hallucinations and delusions at the variant and gene levels within the *DISC1* interactome.

Symptoms	Type of Symptom	SNP (Gene)	Allele	OR (95% CI)	*p*-Value	Gene	Number of SNPs	ZSTAT	*p*-Value
Hallucinations	Auditory hallucinations	rs6754640 (*NRXN1*)	A	2.27 (1.57–3.29)	1.48 × 10^−5^	*NRG1*	30	4.41	5.25 × 10^−6^
						*PCM1*	16	3.32	4.47 × 10^−4^
Delusions	Persecutory delusions	rs17039676 (*NRXN1*)	T	4.89 (2.27–7.52)	8.43 × 10^−4^	*NRG1*	29	3.83	6.35 × 10^−5^
						*APP*	34	3.27	5.34 × 10^−4^
	Delusions of reference	rs6706713 (*NRXN1*)	G	3.40 (2.10–5.50)	6.35 × 10^−7^	*NRXN1*	155	4.33	7.57 × 10^−6^
		rs11892200 (*NRXN1*)	C	3.15 (2.00–4.97)	7.05 × 10^−7^	*EXOC4*	34	3.38	3.69 × 10^−4^
		rs6754640 (*NRXN1*)	A	3.33 (2.02–5.40)	2.54 × 10^−6^	*NUP210*	11	3.55	1.90 × 10^−4^
		rs17039676 (*NRXN1*)	T	4.13 (2.27–7.52)	3.52 × 10^−6^	*APP*	33	3.48	2.52 × 10^−4^
		rs6731061 (*NRXN1*)	T	3.05 (1.90–4.89)	3.95 × 10^−6^				
		rs7578902 (*NRXN1*)	G	2.93 (1.85–4.63)	4.11 × 10^−6^				
		rs10189159 (*NRXN1*)	C	3.14 (1.90–5.20)	7.97 × 10^−6^				
		rs7076156 (*ZNF365*)	A	3.48 (2.00–6.08)	1.13 × 10^−5^				
		rs10263196 (*EXOC4*)	A	3.09 (1.86–5.12)	1.36 × 10^−5^				
		rs10176705 (*NRXN1*)	T	3.11 (1.83–5.28)	2.64 × 10^−5^				
		rs1421579 (*NRXN1*)	G	2.52 (1.63–3.88)	2.83 × 10^−5^				

Amyloid Beta Precursor Protein (*APP*), Exocyst Complex Component 4 (*EXOC4*), Neurexin 1 (*NRXN1*), Neuregulin 1 (*NRG1*), Nucleoporin 210 (*NUP210*), Pericentriolar Material 1 (*PCM1*), Zinc Finger Protein 365 (*ZNF365*).

**Table 2 ijms-26-08738-t002:** Regulatory variants associated with hallucinations and delusions within the DISC1 interactome.

Position (hg19)	Gene	Consequence	SNP	Ref/alt	Freq	GnomAD	Transcript	Effect
chr2:50504180-50504180	*NRXN1*	Downstream gene variant	rs6754640	G/AT	0.1593	0.3127	ENST00000331040.9 ENST00000401669.7	Nonsense-mediated decay Protein coding
chr2:50029801-50029801	*NRXN1*	Intron variant	rs17039676	C/T	0.0766	0.1515	ENST00000637906.1 ENST00000342183.9	Nonsense-mediated decay Protein coding
chr2:50494373-50494373	*NRXN1*	Intron variant	rs6706713	A/G	0.2057	0.3929	ENST00000331040.9 ENST00000401669.7	Nonsense-mediated decay Protein coding
chr2:50480720-50480720	*NRXN1*	Intron variant	rs11892200	T/C	0.2233	0.4258	ENST00000331040.9 ENST00000401669.7	Nonsense-mediated decay Protein coding
chr2:50016264-50016264	*NRXN1*	Intron variant	rs6731061	C/AT	0.1733	0.3410	ENST00000637906.1 ENST00000342183.9	Nonsense-mediated decay Protein coding
chr2:50480256-50480256	*NRXN1*	Intron variant	rs7578902	A/CGT	0.2173	0.4152	ENST00000331040.9 ENST00000401669.7	Nonsense-mediated decay Protein coding
chr2:50487433-50487433	*NRXN1*	Intron variant	rs10189159	T/C	0.1537	0.3058	ENST00000331040.9 ENST00000401669.7	Nonsense-mediated decay Protein coding
chr10:62655424-62655424	*ZNF365*	Non-coding transcript exon variant	rs7076156	A/CGT	0.1002	0.8516	ENST00000344640.7	lncRNA
chr7:133271427-133271427	*EXOC4*	Intron variant	rs10263196	G/A	0.1332	0.2391	ENST00000253861.5	Protein coding
chr2:50517636-50517636	*NRXN1*	Intron variant	rs10176705	C/T	0.1394	0.2706	ENST00000331040.9 ENST00000401669.7	Nonsense-mediated decay Protein coding
chr2:50005007-50005007	*NRXN1*	Intron variant	rs1421579	G/AT	0.3110	0.4547	ENST00000637906.1 ENST00000342183.9	Nonsense-mediated decay Protein coding

Freq = frequency in individuals with psychosis; GnomAD = Genome Aggregation Database; lncRNA = long non-coding RNA; Ref/alt = reference and alternative alleles; SNP = single nucleotide polymorphism.

**Table 3 ijms-26-08738-t003:** Genetic associations with hallucinations and delusions at the gene set level within the *DISC1* interactome.

Symptoms	Type of Symptom	Biological Pathway	Gene	BETA	*p*-Value
Hallucinations	Auditory hallucinations	Non-motile cilia assembly	*PCM1. DISC1*, *BBS4*	1.7	8.01 × 10^−3^
		Cilia organization	*PCM1*, *DISC1*, *BBS4*, *EXOC7*, *KIF3A PCNT*	1.21	1.23 × 10^−2^
		Positive regulation of cellular component organization	*GSK3B*, *NRXN1*, *NRG1*, *CLU*	1.5	1.58 × 10^−2^
		Glutamate receptor signaling pathway	*APP*, *GRIA2*	1.72	2.36 × 10^−2^
		Regulation of amyloid fibril formation	*APP*, *CLU*	1.67	3.29 × 10^−2^
		Regulation of secretion	*KCNQ1*, *NRG1*	1.96	3.30 × 10^−2^
		Microglial cell activation	*APP*, *CLU*	1.67	4.06 × 10^−2^
Delusions	Persecutory delusions	Golgi vesicular transport	*APP*, *EXOC4*, *DTNBP1*	2.3	1.96 × 10^−4^
		Neuronal development	*APP*, *GSK3B*, *CDK5*, *DTNBP1*	2.08	5.86 × 10^−4^

Amyloid Beta Precursor Protein (*APP*), Bardet-Biedl Syndrome 4 (*BBS4*), Clusterin (*CLU*), Cyclin Dependent Kinase 5 (*CDK5*), Disrupted in Schizophrenia 1 (*DISC1*), Dystrobrevin Binding Protein 1 (*DTNBP1*), Exocyst Complex Component 7 (*EXOC7*), Glutamate Ionotropic Receptor AMPA Type Subunit 2 (*GRIA2*), Glycogen Synthase Kinase 3 Beta (*GSK3B*), Kinesin Family Member 3A (*KIF3A*), Neurexin 1 (*NRXN1*), Neuregulin 1 (*NRG1*), Pericentrin (*PCNT*), Pericentriolar Material 1 (*PCM1*), Potassium Voltage-Gated Channel Subfamily Q Member 1 (*KCNQ1*).

## Data Availability

The original data presented in the study are openly available in FigShare at https://doi.org/10.6084/m9.figshare.29991376.

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
