# Peer review of "Differential Association of the *DISC1* Interactome in Hallucinations and Delusions [Author-notes fn1-ijms-26-08738]"

_ijms, 2025, doi:10.3390/ijms26178738_

Round 1
Reviewer 1 Report
Comments and Suggestions for Authors
- The authors ignore negative symptoms when describing the clinical picture of schizophrenia, as well as depressed mood and anhedonia in depression. This is evident in the Abstract (line 20) and 1. Introduction (lines 43-50) sections. As is known, it is the negative symptoms that are the most difficult to treat in schizophrenia and lead to disability more often than hallucinations and delusions. Full-blown hallucinations and delusions are most often encountered in the first episodes of schizophrenia and/or in the acute phase. The mechanism of occurrence of hallucinations and delusions in affective disorders is different. They are a consequence of the severity of the mental state, regardless of their congruence with mood. At the same time, the authors themselves point to the heterogeneity of the psychosis (line 51), referring to the article by Gaebel & Zielasek.
- The hypothesis is missing from the manuscript.
- The authors omitted to mention the mental disorders in which they planned to identify the genetic association between the DISC1 interactome genes and auditory, visual, somatic and olfactory hallucinations, as well as delusions of guilt, grandeur, persecution and thought control in study aim formulation.
- The authors should have divided the sample into two separate groups of patients with schizophrenia and bipolar disorder and tested for differences in the genetic association between the DISC1 interactome genes and auditory, visual, somatic, and olfactory hallucinations, as well as delusions of guilt, reference, grandiosity, persecution and thought control. An alternative would be to provide a compelling rationale for ignoring the categorical approach in the research methodology, drawing on recent evidence.
- Section 4. Materials and Methods does not contain a description of the statistical methods used.
Author Response
For brief report
|
Response to Reviewer 1 Comments |
||
|
1. Summary |
||
|
Thank you very much for taking the time to review this manuscript. Please find the detailed responses below and the corresponding corrections highlighted in red in the re-submitted files. |
||
|
2. Questions for General Evaluation |
Reviewer’s Evaluation |
Response and Revisions |
|
Does the introduction provide sufficient background and include all relevant references? |
Yes/Can be improved/Must be improved/Not applicable |
The text has been revised to include the relevant information. |
|
Are all the cited references relevant to the research? |
Yes/Can be improved/Must be improved/Not applicable |
We included relevant references to provide a better understanding of the study. |
|
Is the research design appropriate? |
Yes/Can be improved/Must be improved/Not applicable |
We described the research design section more clearly. |
|
Are the methods adequately described? |
Yes/Can be improved/Must be improved/Not applicable |
We described the methods section more clearly. |
|
Are the results clearly presented? |
Yes/Can be improved/Must be improved/Not applicable |
We revised the tables and also added a figure |
|
Are the conclusions supported by the results? |
Yes/Can be improved/Must be improved/Not applicable |
We provided a more in-depth analysis of the findings. |
3. Point-by-point response to Comments and Suggestions for Authors |
||
|
Comments 1: The authors ignore negative symptoms when describing the clinical picture of schizophrenia, as well as depressed mood and anhedonia in depression. This is evident in the Abstract (line 20) and 1. Introduction (lines 43-50) sections. As is known, it is the negative symptoms that are the most difficult to treat in schizophrenia and lead to disability more often than hallucinations and delusions. Full-blown hallucinations and delusions are most often encountered in the first episodes of schizophrenia and/or in the acute phase. The mechanism of occurrence of hallucinations and delusions in affective disorders is different. They are a consequence of the severity of the mental state, regardless of their congruence with mood. At the same time, the authors themselves point to the heterogeneity of the psychosis (line 51), referring to the article by Gaebel & Zielasek. |
||
|
Response 1: Thank you very much for your comments. We have mentioned the negative symptoms in schizophrenia (Sameer Jauhar et al., 2022, DOI: 10.1016/S0140-6736(21)01730-X); however, we did not explore them in depth, as this is beyond the focus of the article. In addition, negative or affective symptoms were not considered in this study because the bipolar disorder group analyzed was not limited to individuals exhibiting only affective symptoms, but rather included those with a history of psychosis. Given this, and since evidence indicates that schizophrenia and bipolar disorder share a common clinical and genetic basis (Cardo & Owen, 2014, https://doi.org/10.1093/schbul/sbu016), we focused solely on describing psychosis in both disorders and the most characteristic symptoms observed in our population. Furthermore, major depressive disorder was excluded, as it is not relevant to the study and could lead to potential misinterpretation. We also revised the statements on heterogeneity for better comprehension. See Introduction, lines 20 and 44–50. |
||
|
Comments 2: The hypothesis is missing from the manuscript. |
||
|
Response 2: Thank you for pointing this out; we have included the study hypothesis in the manuscript. See lines 60–62. |
||
|
Comments 3: The authors omitted to mention the mental disorders in which they planned to identify the genetic association between the DISC1 interactome genes and auditory, visual, somatic and olfactory hallucinations, as well as delusions of guilt, grandeur, persecution and thought control in study aim formulation. |
||
|
Response 3: Thank you for mentioning this; we have added the specific diagnoses to be studied in the objective. See lines 67-68. |
||
|
Comments 4: Section 4. Materials and Methods does not contain a description of the statistical methods used. |
||
|
Response 4: We have mentioned the statistical analyses in the manuscript, including SNP-level association using PLINK, gene- and gene-set level analyses using MAGMA, and multiple testing correction with the False Discovery Rate (FDR) in R. The text has also been rewritten for improved clarity. See lines 267-293.. |
|
4. Response to Comments on the Quality of English Language |
||
|
Point 1: |
||
|
Response 1: The manuscript has been carefully revised to improve clarity and readability in English. |
- Additional clarifications
The tables were revised to improve clarity and comprehension, and an additional figure was included to better illustrate the findings.
Reviewer 2 Report
Comments and Suggestions for Authors
This study put up a new perspective by focusing on symptom-specific genetic associations within the DISC1 interactome, particularly distinguishing among different types of hallucinations and delusions. While prior research has largely emphasized global psychosis phenotypes or broad diagnostic categories, this work refines the approach by mapping genetic risk at the level of discrete symptoms. This granularity represents a valuable contribution to psychiatric genetics, especially in light of the growing recognition of symptom-level heterogeneity in disorders such as schizophrenia and bipolar disorder.
It is commendable that the authors combined SNP-, gene-, and gene set-level analyses, and conducted the study in a Mexican population, which is often underrepresented in genomic research. The rigorous quality control procedures, use of MAGMA and PLINK, and inclusion of substance-use covariates add robustness to the findings. The identification of genes such as NRXN1, EXOC4, and NRG1 as associated with specific symptom dimensions (e.g., auditory hallucinations, persecutory delusions) provides mechanistic insight that could inform future diagnostic or therapeutic strategies.
Overall, this work provides meaningful insights into the genetic architecture of psychotic symptoms and represents a valuable addition to the field. However, there may be several issues in the article that require further revision.
1. Confidence intervals in Table 1 appear inconsistent with reported odds ratios. For example, an OR of 4.89 with a CI of (0.36–2.43) is mathematically invalid. Please re-calculate and verify all ORs and CIs to ensure that the lower and upper bounds are consistent with the reported effect size.
2. References [18] and [19] appear to be duplicates, citing the same study.
3. It is recommended to include a pathway enrichment plot or a schematic diagram of key DISC1-related genes to present the findings more intuitively.
4. The manuscript reports strong associations at the gene level (e.g., NRXN1, EXOC4) but no significant findings at the gene set level for some symptoms (e.g., delusions of reference). This discrepancy should be addressed and discussed in the manuscript.
5. Line 140 contains a grammatical error in the phrase “which allow determine endophenotypes...”.
6. Although the introduction has background, it does not highlight "what problems this study aims to solve"
7. " Psychosis has a high heritability (~80%), and genetic association studies of various psychiatric disorders with psychosis... have shown overlapping risk loci associated with this phenotype." The logical subject is unclear. Who exactly has overlapping risk loci? Is it disorders or psychosis?
Author Response
For brief report
|
Response to Reviewer 2 Comments |
||
|
1. Summary |
||
|
Thank you very much for taking the time to review this manuscript. Please find the detailed responses below and the corresponding corrections highlighted in red in the re-submitted files. |
||
|
2. Questions for General Evaluation |
Reviewer’s Evaluation |
Response and Revisions |
|
Does the introduction provide sufficient background and include all relevant references? |
Yes/Can be improved/Must be improved/Not applicable |
The text has been revised to include the relevant information. |
|
Are all the cited references relevant to the research? |
Yes/Can be improved/Must be improved/Not applicable |
We included relevant references to provide a better understanding of the study. |
|
Is the research design appropriate? |
Yes/Can be improved/Must be improved/Not applicable |
We described the research design section more clearly. |
|
Are the methods adequately described? |
Yes/Can be improved/Must be improved/Not applicable |
We described the methods section more clearly. |
|
Are the results clearly presented? |
Yes/Can be improved/Must be improved/Not applicable |
We revised the tables and also added a figure |
|
Are the conclusions supported by the results? |
Yes/Can be improved/Must be improved/Not applicable |
We provided a more in-depth analysis of the findings. |
3. Point-by-point response to Comments and Suggestions for Authors |
||
|
Comments 1: This study put up a new perspective by focusing on symptom-specific genetic associations within the DISC1 interactome, particularly distinguishing among different types of hallucinations and delusions. While prior research has largely emphasized global psychosis phenotypes or broad diagnostic categories, this work refines the approach by mapping genetic risk at the level of discrete symptoms. This granularity represents a valuable contribution to psychiatric genetics, especially in light of the growing recognition of symptom-level heterogeneity in disorders such as schizophrenia and bipolar disorder. It is commendable that the authors combined SNP-, gene-, and gene set-level analyses, and conducted the study in a Mexican population, which is often underrepresented in genomic research. The rigorous quality control procedures, use of MAGMA and PLINK, and inclusion of substance-use covariates add robustness to the findings. The identification of genes such as NRXN1, EXOC4, and NRG1 as associated with specific symptom dimensions (e.g., auditory hallucinations, persecutory delusions) provides mechanistic insight that could inform future diagnostic or therapeutic strategies. Overall, this work provides meaningful insights into the genetic architecture of psychotic symptoms and represents a valuable addition to the field. However, there may be several issues in the article that require further revision. Thank you very much for your valuable comments on our work. 1. Confidence intervals in Table 1 appear inconsistent with reported odds ratios. For example, an OR of 4.89 with a CI of (0.36–2.43) is mathematically invalid. Please re-calculate and verify all ORs and CIs to ensure that the lower and upper bounds are consistent with the reported effect size. |
||
|
Response 1: Thank you for pointing this out; an error occurred during the transfer of that data to the tables. It should read OR (95% CI) 4.89 (2.27–7.52). We have verified that the remaining data is correct. |
||
|
Comments 2: References [18] and [19] appear to be duplicates, citing the same study. |
||
|
Response 2: We appreciate your observation. The duplicate reference has been removed, and the remaining references have been carefully reviewed to ensure accuracy and absence of duplication. |
||
|
Comments 3: It is recommended to include a pathway enrichment plot or a schematic diagram of key DISC1-related genes to present the findings more intuitively. |
||
|
Response 3: Thank you for your suggestion. A pathway enrichment diagram has been included to illustrate the association at the gene set level. |
||
|
Comments 4: The manuscript reports strong associations at the gene level (e.g., NRXN1, EXOC4) but no significant findings at the gene set level for some symptoms (e.g., delusions of reference). This discrepancy should be addressed and discussed in the manuscript. |
||
|
Response 4: We appreciate the comment. A discussion addressing the discrepancy in these results has been added. See lines 219-231. |
||
Comments 5: Line 140 contains a grammatical error in the phrase “which allow determine endophenotypes...”. |
||
|
Response 5: The grammatical error has been corrected. See lines 165-167. |
||
Comments 6: Although the introduction has background, it does not highlight "what problems this study aims to solve" |
||
|
Response 6: The problem our study aims to solve has now been described in the Introduction section. See lines 55-60. |
||
|
Comments 7: " Psychosis has a high heritability (~80%), and genetic association studies of various psychiatric disorders with psychosis... have shown overlapping risk loci associated with this phenotype." The logical subject is unclear. Who exactly has overlapping risk loci? Is it disorders or psychosis? |
||
|
Response 7: The statement has been rewritten to clarify the genetic overlap. |
|
4. Response to Comments on the Quality of English Language |
||
|
Point 1: The English could be improved to more clearly express the research. |
||
|
Response 1: The manuscript has been carefully revised to improve clarity and readability in English. |
Round 2
Reviewer 1 Report
Comments and Suggestions for Authors
The authors have done a great job of improving the manuscript. I thank the authors for the detailed answer regarding the sample formation and accept the logic and all the arguments made, except for the statistical power. It would have been possible to take more time to form two samples, which would have made the article even more interesting.
Author Response
For article
|
Response to Reviewer 1 Comments |
||
|
1. Summary |
||
|
Thank you very much for taking the time to review this manuscript. Please find the detailed responses below. |
||
|
2. Questions for General Evaluation |
Reviewer’s Evaluation |
Response and Revisions |
|
Does the introduction provide sufficient background and include all relevant references? |
Yes/Can be improved/Must be improved/Not applicable |
Thank you so much for your careful review and valuable feedback. We truly appreciate your time and insights. |
|
Are all the cited references relevant to the research? |
Yes/Can be improved/Must be improved/Not applicable |
|
|
Is the research design appropriate? |
Yes/Can be improved/Must be improved/Not applicable |
|
|
Are the methods adequately described? |
Yes/Can be improved/Must be improved/Not applicable |
|
|
Are the results clearly presented? |
Yes/Can be improved/Must be improved/Not applicable |
|
|
Are the conclusions supported by the results? |
Yes/Can be improved/Must be improved/Not applicable |
|
3. Point-by-point response to Comments and Suggestions for Authors |
||
|
Comments 1: The authors have done a great job of improving the manuscript. I thank the authors for the detailed answer regarding the sample formation and accept the logic and all the arguments made, except for the statistical power. It would have been possible to take more time to form two samples, which would have made the article even more interesting. Thank you so much for your careful review and valuable feedback. We truly appreciate your time and insights. We will definitely take your comments regarding the division of study groups into account for future work. Additionally, in this revised manuscript, we have incorporated a new table (Table 2) and figure (Figure 2) to improve the clarity of the results. We have also updated the literature review to include recent studies that jointly analyze schizophrenia and bipolar disorder to identify shared genetic risk variants. |
||
Reviewer 2 Report
Comments and Suggestions for Authors
- The table should be changed as the three line style.
- The list of the abbreviations do not include the related description, such as GWAS, KEGG and etc.
- Line 147: wrong format.
- In the method, the author should separate as several parts by subtitles. What's more, author mentioned that GO analysis in functional analysis, but I can’t find the related results.
- The raw data supporting the conclusions of this article should be uploaded to the public dataset.
- In this article, the author have cited 31 references,and few researh have been published in recent 5 years. The author need to refresh the the references.
- The author have provide several SNPs that related the DISC1 interactome. The author should provide more details, such as whether these SNPs have influenced the amino acid sequence or protein structure.
- In this manuscript,the author only use 2 tables and 1 figures in the results. To support the conclusion, the author still need more consructive figures.
Author Response
For article
|
Response to Reviewer 2 Comments |
||
|
1. Summary |
||
|
Thank you very much for taking the time to review this manuscript. Please find the detailed responses below and the corresponding corrections highlighted in red in the re-submitted files. |
||
|
2. Questions for General Evaluation |
Reviewer’s Evaluation |
Response and Revisions |
|
Does the introduction provide sufficient background and include all relevant references? |
Yes/Can be improved/Must be improved/Not applicable |
The text has been revised to include the relevant information. |
|
Are all the cited references relevant to the research? |
Yes/Can be improved/Must be improved/Not applicable |
We included more recent references to provide a better understanding of the study. |
|
Is the research design appropriate? |
Yes/Can be improved/Must be improved/Not applicable |
We described the research design section more clearly. |
|
Are the methods adequately described? |
Yes/Can be improved/Must be improved/Not applicable |
We described the methods section more clearly. |
|
Are the results clearly presented? |
Yes/Can be improved/Must be improved/Not applicable |
We revised the tables and also added a table and a figure. |
|
Are the conclusions supported by the results? |
Yes/Can be improved/Must be improved/Not applicable |
We provided a more in-depth analysis of the findings. |
3. Point-by-point response to Comments and Suggestions for Authors |
||
|
Comments 1: The table should be changed as the three line style. |
||
|
Response 1: All tables have been formatted according to the three-line style. |
||
|
Comments 2: The list of the abbreviations do not include the related description, such as GWAS, KEGG and etc. |
||
|
Response 2: The descriptions of the abbreviations used in the study have been included. See lines 50–51, 112–113, 178–179, 353–356, and the list of abbreviations on line 394. |
||
|
Comments 3: Line 147: wrong format. |
||
|
Response 3: Thank you for pointing this out. The error has been corrected. See line 157. |
||
|
Comments 4: In the method, the author should separate as several parts by subtitles. What's more, author mentioned that GO analysis in functional analysis, but I can’t find the related results. |
||
|
Response 4: The Methods section has been divided into segments. See the Materials and Methods section, lines 306–363. The pathway enrichment analysis using Gene Ontology (GO) has been presented more clearly in the text, as the results from this analysis were not previously highlighted. See lines 112–114, 154 (Figure 1), 161 (Figure 2), and 352–356. |
||
Comments 5: The raw data supporting the conclusions of this article should be uploaded to the public dataset. |
||
|
Response 5: The original data presented in the study are openly available in FigShare at 10.6084/m9.figshare.29991376. |
||
Comments 6: In this article, the author have cited 31 references,and few researh have been published in recent 5 years. The author need to refresh the the references. |
||
|
Response 6: We appreciate your observation. We have updated the references, incorporating recent studies from 2022 to 2025 that provide valuable and current insights, enhancing the discussion of our study. Friligkou E, et al., 2025, Psychol Med, doi:10.1017/S0033291725001217; Nadella RK, et al., 2023, Indian J Psychol Med, doi:10.1177/02537176221084867; Trubetskoy V, et al., 2022, Nature, doi:10.1038/s41586-022-04434-5; Wei Y, et al., 2023, Biol Psychiatry, doi:10.1016/j.biopsych.2022.11.006; Mikhalitskaya EV, et al., 2023, Genes, doi:10.3390/genes14071460. |
||
|
Comments 7: The author have provide several SNPs that related the DISC1 interactome. The author should provide more details, such as whether these SNPs have influenced the amino acid sequence or protein structure. |
||
|
Response 7: Thank you for the suggestion. We have added an analysis of the SNPs to identify regulatory variants using the Variant Effect Predictor (VEP). See Table 2 (lines 134–143), lines 182–187, and lines 343–345. |
|
Comments 8: In this manuscript,the author only use 2 tables and 1 figures in the results. To support the conclusion, the author still need more consructive figures. |
||
|
Response 8: We have included an additional table (Table 2) illustrating the regulatory functions of the SNPs, along with a figure (Figure 2) showing the biological pathway networks from the gene-set analysis, taking into account the genes involved in each pathway. See lines 134 and 161. |
|
4. Response to Comments on the Quality of English Language |
||
|
Point: The English could be improved to more clearly express the research. |
||
|
Response: The manuscript has been carefully revised to improve clarity and readability in English. We have identified and corrected errors in both grammar and writing. |
- Response to Comments on the Quality of Figures
Point: Figures and tables can be improved
Response: We have added a new table (Table 2) and a figure (Figure 2) to illustrate the effect of the genetic variants, as well as the gene network with associated biological pathways. Additionally, we have revised the format of the existing tables to improve the clarity and readability of the results.
Round 3
Reviewer 2 Report
Comments and Suggestions for Authors
LINE 390: The abbreviation Table should list all the mentioned abbreviations in this manucript, including the gene symbols。
Author Response
For article
|
Response to Reviewer 2 Comments |
||
|
1. Summary |
||
|
Thank you very much for taking the time to review this manuscript. Please find the detailed responses below and the corresponding corrections highlighted in red in the re-submitted files. |
||
|
2. Questions for General Evaluation |
Reviewer’s Evaluation |
Response and Revisions |
|
Does the introduction provide sufficient background and include all relevant references? |
Yes/Can be improved/Must be improved/Not applicable |
Thank you so much for your careful review and valuable feedback. We truly appreciate your time and insights. |
|
Are all the cited references relevant to the research? |
Yes/Can be improved/Must be improved/Not applicable |
|
|
Is the research design appropriate? |
Yes/Can be improved/Must be improved/Not applicable |
|
|
Are the methods adequately described? |
Yes/Can be improved/Must be improved/Not applicable |
|
|
Are the results clearly presented? |
Yes/Can be improved/Must be improved/Not applicable |
|
|
Are the conclusions supported by the results? |
Yes/Can be improved/Must be improved/Not applicable |
|
|
Are all figures and tables clear and well-presented? |
Yes/Can be improved/Must be improved/Not applicable |
We have reviewed the table mentioned in your comments. |
3. Point-by-point response to Comments and Suggestions for Authors |
||
|
Comments 1: LINE 390: The abbreviation Table should list all the mentioned abbreviations in this manucript, including the gene symbols。 |
||
|
Response 1: Thank you for your valuable feedback. We have revised the table to include the abbreviations used in the manuscript. See Abbreviations section, line 390. |
||